# A Review of the Effect of Formic Acid and Its Salts on the Gastrointestinal Microbiota and Performance of Pigs

**DOI:** 10.3390/ani10050887

**Published:** 2020-05-19

**Authors:** Diana Luise, Federico Correa, Paolo Bosi, Paolo Trevisi

**Affiliations:** Department of Agricultural and Food Sciences (DISTAL), University of Bologna, 40127 Bologna, Italy; federico.correa2@unibo.it (F.C.); paolo.bosi@unibo.it (P.B.); paolo.trevisi@unibo.it (P.T.)

**Keywords:** microbiota, formic acid, pig, sow, organic acid

## Abstract

**Simple Summary:**

Acidifiers, especially formic acid and its salts, have long been used for improving the gastrointestinal health and growth performance of pigs, especially piglets. Although, some biological mechanisms explaining their effects are already known, a lack of consensus is still reported on their effects on the gastrointestinal microbiome. The present review provides an overview and evaluation of the effects of formic acid and its salts on the gastrointestinal microbiota and on the growth performance of pigs.

**Abstract:**

Out of the alternatives to antibiotics and zinc oxide, organic acids, or simply acidifiers, play significant roles, especially in ensuring gut health and the growth performance of pigs. Regarding acidifiers, formic acid and its salts have shown very promising results in weaning, growing and finishing pigs. Although it is known that the main mechanisms by which acidifiers can improve livestock performance and health are related to the regulation of gastrointestinal pH, an improvement in intestinal digestibility and mineral utilization, and their antimicrobial properties against specific pathogens has been observed, while poor consensus remains in relation to the effect of acidifers on bacteria and the complex microbiome. Therefore, the aim of the present review was to critically evaluate the effects of formic acid and its salts on the performance and the gastrointestinal microbiota balance of pigs.

## 1. Introduction

An increase in the prevalence of antibiotic-resistant bacteria has led to growing discussion and concern regarding the proper use of antimicrobial substances in animal feed [1], as well as the supranutritional supplementation (2500–3000 ppm) of zinc oxide (ZnO) as the most common alternatives to antibiotics. Therefore, the search for alternatives to antibiotics and ZnO has aroused common interest in the feed industry, and for farmers and researchers. Of the different potential strategies, organic acids and their salts have been studied extensively, showing promising results in controlling intestinal disease and improving animal growth performance, especially in pigs and poultry [2,3,4].

Acidifiers have been widely used for decades in livestock feeds, due to their preservative and nutritional qualities [4,5]. Initially, they were used to control the proliferation of bacteria, fungi or molds in feedstuffs for livestock [6]. More recently, a number of studies have highlighted the potential effects of organic acids in improving digestion, nutrient digestibility, intestinal health and the promotion of growth performance in monogastric animals, as well as the sanitizing effect that they have on the feed [7,8,9,10]. Of the different organic acids, formic acid and its derivatives (ammonium formate, K-diformate, Ca-formate, Na-formate) have been intensively studied in pigs with very promising results in weaning [11,12,13], growing [14,15] and finishing pigs [14,16,17]. However, little is known about the main biological mechanism on the effects of formic acid on the growth and health of pigs. 

It is known that the main mechanisms by which organic acids can improve livestock growth performance are related to an improvement in intestinal digestibility and mineral utilization, and to their antimicrobials properties regarding specific pathogens [18,19,20,21]. However, little information regarding the effect of organic acids on hindgut eubiosis is available. Throughout the past ten years, the advent and improvements in Next Generation Sequencing (NGS) technologies have stimulated metagenomics research, increasing the capability of simultaneously analyzing communities of microorganisms [22]. The in-depth description of the microbial profile using the NGS approach can therefore contribute to explaining the potential effect of organic acids on microbial composition and the gut health of livestock. Therefore, the present review provides an overview and evaluation regarding the effects of formic acid and its salts on the gastrointestinal microbiota considering *in vitro* and *in vivo* studies, and on the performance of pigs.

## 2. Physicochemical Properties, Production and Use of Formic Acid and its Salts

Acids, like all chemical compounds, are classified as either organic or inorganic. Organic acids are characterized by a hydrogen and a carbon, and at least one other element, resulting in the presence of the carboxyl functional group (-COOH or -CO_2_H). In general, the organic acids receiving most attention have been the short-chain fatty acids (SCFAs) due to their suitability as feed preservatives and their specific antimicrobial activity. SCFA are composed of the R-COOH structure in the form of a monocarboxylic acid (formic, acetic, propionic and butyric acid), as a hydroxyl group (lactic, malic, tartaric and citric acids) or as a double bond (fumaric and sorbic acids) [23]. Organic acids are mainly classified according to length as SCFAs characterized by an aliphatic tail of fewer than six carbons (i.e., butyric acid and formic acid), medium-chain fatty acids (MCFAs) with aliphatic tails of 6–12 carbons which constitute the medium-chain triglycerides, long-chain fatty acids (LCFAs) with aliphatic tails of 13 to 21 carbons and very long chain fatty acids (VLCFAs) with aliphatic tails longer than 22 carbons.

An acid is a substance which yields an excess of hydrogen ions when dissolved in water. Some acids are stronger than others. The relative acidity of different compounds (in other words, their relative capacity to donate a proton to a common base under identical conditions) is quantified by a number called the dissociation constant, abbreviated by Ka, and is sometimes also expressed on a logarithmic scale as pKa values:Ka = (H_3_O) × (A^−^)/(HA)pKa = − logKa = log (1/Ka).(1)

The relationship between solution acidity (pH), the dissociation constant (pKa), and the relative concentrations of acids is as follows,
pH = pKa + log (A^−^)/(HA)(2)
where (A^−^) is the molar concentration of the dissociated salt and (HA) is the concentration of the undissociated acid. When the concentrations of the salt and the acid are equal, the pH of the system equals the pKa of the acid. Thus, the lower the pKa value, the stronger the acid. Solvent, temperature and concentrations of reactants are the main factors influencing the reactivity of SCFAs. Reactivity is the tendency of a substance to undergo a chemical ionic reaction, either by itself or with other materials. The reactivity of members of a particular class of carbonyl compounds depends upon the steric and electronic environment of the carbonyl group. Hydrocarbons that do not contain a benzene ring are known as aliphatic hydrocarbons. Those which do contain a benzene ring are called aromatic hydrocarbons. In general, aromatic compounds are less reactive than their aliphatic counterparts because the partial positive charge on the carbonyl carbon atom can be delocalized around the benzene ring. The majority of SCFAs and their salts exhibit appreciable solubility in water due to the presence of the polar carboxyl group, although this solubility declines with an increasing number of C-atoms in the molecule [24]. The physiochemical properties of some acids and salts are reported in Table 1.

### 2.1. Physicochemical Properties and the Production of Formic Acid and its Salts

Formic acid (H_2_CO_2_ or HCOOH), is the first member of the series of aliphatic monobasic acids having the general formula CH_2_O_2_. It is a colorless liquid, soluble in water and alcohol, with a pungent odor, a molecular weight of 46.03 g/mol, a density of 1.22 g/cm^3^ at 20 °C and a pKa of 3.75 at 20 °C (Table 1). It has stronger acidity and corrosive properties than other fatty acids (FAs), 

It can be obtained artificially using several processes, including the oxidation of methyl alcohol and of formaldehyde, the rapid heating of oxalic acid and the hydrolysis of hydrocyanic acid by means of hydrochloric acid, but mainly by heating oxalic acid with glycerin [25]. 

In addition to the free-form, the salification of formic acid is a well-known process. Formate is a monocarboxylic acid anion which is the conjugate base of formic acid (CHO_2_(−)) with a molecular weight of 45.017 g/mol; it is produced by the action of moist carbon monoxide on soda lime or by the action of moist carbon dioxide on potassium. 

Potassium formate (CHKO_2_, molecular weight 84.116 g/mol) and potassium diformate (C_2_H_3_O_4_K, molecular weight 130.12 g/mol) are salts derived from formic acid and potassium, and authorization as a feed additive by the European Food Safety Authority (EFSA) has been provided only for potassium diformate [26]. Potassium formate can be obtained from chloroform boiled with alcoholic potash; a somewhat similar decomposition has been shown by chloral and aqueous potash. Potassium diformate is produced from equimolar amounts of formic acid and potassium formate, separated by centrifugation into a crystalline mass of potassium diformate and a saturated aqueous solution of potassium diformate, which is diluted with water up to 45–55%. 

Sodium formate (HCOONa, molecular weight 68.008 g/mol) in its solid and liquid forms is recognized as a technological feed additive by EFSA [27,28]. Solid sodium formate is generally obtained as a by-product in the production of pentaerythritol, trimethylolpropane or neopentyl glycol. One of the starting materials (acetaldehyde, n-butyraldehyde or isobutyraldehyde) is combined with formaldehyde and sodium hydroxide. The resulting sodium formate is isolated by crystallization. The liquid sodium formate is produced by mixing an aqueous solution of sodium hydroxide (50% w/w) with formic acid.

Calcium formate (HCOO_2_Ca; molecular weight 130.1 g/mol) derives from the reaction between dimethylol butyraldehyde (DIMBA) and formaldehyde, which produces trimethylolpropane (TMP) and calcium formate. The calcium formate is then separated from the TMP-calcium formate solution, heated to remove the formaldehyde and then dried [25].

Ammonium formate (HCOONH_4,_ molecular weight 63.06 g/mol) is another salt, which can be used as an additive in animal feeding. The addition of gaseous ammonia, or an aqueous solution of ammonium hydroxide to formic acid is used to produce this salt, followed by partial neutralization [28]. This generates a mixture of approximately 35% ammonium formate and almost all the remaining quantity is free formic acid [28]. 

### 2.2. Safety Regarding the Use of Formic Acid and Its Salts in Pig Feed

Formic acid is physiologically produced by the animal metabolism. It does not affect growth performance negatively up to 12 or 18 g/kg feed respectively [15,29] in piglet diets, and up to 18 g/kg feed in growing pigs [17]. The use of formic acid and its salts is, in general, permitted in the animal feeding legislations. In the European Union, formic acid and sodium formate are approved as feed preservatives, as hygiene condition enhancers and also as silage additives [30]. Formic acid is also approved as a flavoring agent, while calcium formate and ammonium formate are approved as feed preservatives. The maximum value of formic acid approved for the use in pigs in the European Union (EU) is 12 g/kg of a feed with a 12 % moisture.

Formamide develops in the production of ammonium formate and/or during its storage. The production process can limit this value to 0.3% of the additive. However, this has raised concerns because formamide in rabbits, mice and rats has been shown to cause embryotoxicity and teratogenesis after oral gavage [31]. For this reason, the EFSA Panel on Additives and Products or Substances used in Animal Feed (FEEDAP) has advised that ammonium formate should not be fed to animals reared for reproductive purposes [28,32]. 

### 2.3. Practices for Formic Acid Inclusion in Feed

Formic acid and its salts can generally be used in different forms: Free or microencapsulated, singly or in blends of acids. The response to mixed acids is usually better than to single acids, perhaps due to the diverse pKa of the acids, which make them effective at different sites of the pig’s digestive tract [33,34]. 

Microencapsulation allows avoiding a fast dissociation of the acid in the stomach. This permits the acids to exert their effects on the distal parts of the intestine, where the largest part of the pathogens, including Salmonella and *Escherichia coli* (*E. coli)*, are present [35,36]. Depending on the susceptibility of the active ingredients to the specific coating methods, microencapsulation can be carried out in two different ways: Chemically or mechanically [24]. 

However, in recent years, researchers have extended their interest to formic acid salts containing calcium (Ca), sodium (Na) or potassium (K). The advantage of salts over free acids is that they are generally odorless and easier to handle in the feed manufacturing process due to their solid and less volatile form. Moreover, the salinized forms are less irritant to the eyes and skin and, in some cases, may be more soluble in water than free acids [4]. In effect, with their improved handling characteristics and better palatability to livestock, formates have been found to be a more user-friendly method of applying formic acid to feed and water, without compromising its efficacy. 

It is also possible to add the acidifiers to drinking water. This is a more valid strategy in weaning piglets because they start drinking more rapidly than eating after weaning. De Busser et al. [37] tested the addition of organic acid to the drinking water of weaning piglets, reducing the water pH to 6, 5 or 4. The authors demonstrated that pH 4 water reduced the fecal excretion of *E. coli* and improved the piglet’s health. Furthermore, in pigs, adding an organic acid-based blend to the drinking water (reducing the pH to 4.6) increased the average daily gain (ADG) by 25 g/day between days 21 and 34 post-weaning [38]. Nevertheless, adding acidifiers to the drinking water could have some disadvantages. First, drinking nipples have to be checked regularly to control the growth of fungal and iron precipitation. Second, the strong acidification of the water might decrease the water intake [37,38]. 

## 3. The Effect of Formic Acid and Its Salts on the Gut Microbial Profile

The hindgut microbiota is known to have a symbiotic relationship with the gut mucosa [39], enhancing the stimulation of the immune system development and imparting substantial metabolic functions of the host [40,41], which could influence the host growth performance as demonstrated in pigs in different stages of life [42,43,44]. The gut microbiota profile and structure can be influenced by different factors, including genetics [45,46], perturbation of specific bacteria [42,47], age and maturation of the host [48,49], diet [50] and specific intestinal environmental conditions, such as gut pH. Of these factors, diet represents one of the main factors capable of rapidly changing the gut-related microbial profile [50]. 

The main target site in the digestive system for free formic acid, and its salts, is the stomach. The gastric environment plays a curtail role for the bacteria selection and acts as an ecological filter, thus, it is fundamental in shaping the structure of the whole gut microbial community [51]. 

Formic acid and its salts exert indirect and direct actions on the gastric microbial population that can affect the microbial composition in the next tracts of the intestine. The main mechanisms by which organic acids can influence the intestinal microbiota are; (1) indirect, by reducing the pH of the environment; and (2) direct by penetrating the bacterial cell and altering its physiological homeostasis. 

A reduction in the intestinal pH leads to the modification of the intestinal microbial composition, reducing or eliminating the pathogenic bacteria, usually sensitive to a low pH, and selecting the acid-resistant microbes, such as lactic acid bacteria [20]. 

The role of formic acid and its salts in reducing the pH has been well documented by Bolduan et al. [52] and Maribo et al. [53] in whose studies the supplementation at different doses of formic acid (0.35% or 1.2%, and 1.4% in combination with lactic acid, respectively) to the piglet diet lowered the gastric and hindgut pH [52,53]. Moreover, Mroz et al., [20] observed a reduced pH in the duodenal digesta of piglets up to 65 hours post-feeding with 0.9% and 1.8% potassium diformate. On the other hand, Roth et al. [54] observed an increase in pH values in the contents of the small and large intestines of weaned piglets fed with 0.6%, 1.2%, 1.8% or 2.4% of formic acid. However, as observed by Canibe et al. [17], the pH in the gastrointestinal tract can vary according to time after feeding at which samples are taken, thus, the different results among the studies could be due to the effect of time between feeding and slaughtering. 

Less is known about the direct effect of formic acid and its salts on the bacterial cells in the intestinal tracts of animals. The direct effect of organic acids on the bacterial cells is related to their capacity to penetrate across the cell membrane of the bacteria. Once in the bacterial cell, where the pH is 7, the formic acid dissociates and decreases the cytoplasm pH producing the inactivation of bacterial decarboxylases and catalases. Consequently, to recompense the balance and restore the cytoplasmic pH to normal conditions, the cell is forced to use energy to expel protons out across the membrane via H+-ATPase. The protein expulsion leads to an accumulation of acid anions in the cell which inhibits intracellular metabolic reactions, including the synthesis of macromolecules, slows down their growth and disrupts internal membranes [24,55,56]. 

The capacity of acids to penetrate the bacterial cells is dependent on their pKa because they can penetrate the surrounding the bacteria in their undissociated form. Formic acid has a pKa of 3.74; thus, a large part of it dissociates in the stomach of post-weaning piglets, since the stomach pH is higher than 3.74. Subsequently, a low concentration of formic acid remains undissociated to exert a direct effect on the bacterial cells. On the other hand, in sows and adult pigs, the stomach pH is relatively low (2.5–4) [57]. Therefore, a higher concentration of formic acid remains undissociated and can exert a direct effect on the bacterial cells. 

The effect of formic acid and its salts on the microbial profile of the hindguts of pigs has still to be elucidated since only a few studies using the NGS approach have been carried out [8,10,58,59,60]. However, it has been well-established that formic acid and its salts have antimicrobial properties, and their effects on the bacterial composition have been intensively studied in pigs, using a culture-based approach applied to both, *in vitro* and *in vivo* studies. 

### 3.1. In Vitro Studies

The antimicrobial properties of formic acid and its salts has been studied in a limited number of outdated *in vitro* studies [61,62,63]. The minimum inhibitory concentrations (MICs), the specific bacterial count, including coliform bacteria, lactobacilli and Salmonella, and gas production have been used as parameters. 

The MIC represents the lowest dose of an organic acid sufficient to inhibit the growth and the multiplication of a microorganism. Formic acid can be more effective in controlling the growth of different classes of pathogens, resulting in lower MIC values than propionic and lactic acids. Formic acid has a MIC value of 0.1 for *S.typhimurium, L.monocytogenes, C. jejuni, Cl. Perfringens* and of 0.15 for *E. coli* and *S.aureus* as compared to an average MIC value of 0.21 for propionic acid and of 0.32 for lactic acid [64]. Moreover, no additive effects were observed when formic acid was mixed with other organic acids [64].

Considering the effect of formic acid on the specific bacterial count, it has been established that formic acid is able to reduce the growth of coliform and of *Salmonella Typhimurium*, but not of lactobacilli, in the gastric digesta, being more effective than other organic acids with similar pKa, such as butyric (pKa 4.82), lactic (pKa 3.86) and fumaric (pKa 3.55) acids [61,62,65]. Furthermore, the authors showed that the effect of organic acids on bacterial growth depended on the pH of the gastric content, and that the effect of the organic acids improved at a lower pH [61,62,65].

### 3.2. In Vivo Studies

The antimicrobial properties of organic acids depend on several factors, and translating the results obtained from the *in vitro* data into *in vivo* could be affected by the ability of the model to mime the GIT environment. Therefore, several *in vivo* studies have been carried out to correctly assess the effect of formic acid supplementation to the feed on the gut microbiota and bacterial counts of pigs. However, comparing the studies poses problems due to the heterogeneity of the experimental protocol (GIT sampling, formic acid concentration and/or type of formic acid, feed intake registration, etc.). Table 2 summarizes the effect of formic acid and its salts on the intestinal bacterial counts of weaned pigs. 

Maribo et al. [53] observed a reduction in lactobacilli in the distal small intestine and cecum, coliforms in the stomach and yeast along the GIT with a supplementation of 0.7% or 1.4% of formic acid in the piglet diet, independent of the dose. In accordance with Maribo et al. [53], Gedek et al. [66] observed a reduction in lactobacilli in the small and large intestines, and a reduction in coliforms in the large intestine, regardless of the dose of formic acid (1.2% to 2.4%), while the level of coliforms in the small intestine was dose-dependent.

The mix of lactic and formic acids led to a reduction in lactobacilli in the stomach without significant alteration of the pH [72,75], while controversial results were observed for the Enterobacterial level, which was reduced only in the study of Hansen et al. [72]. As an alternative to free formic acid, several studies have tested the addition of potassium diformate because of its strong antimicrobial activity. Fevrier et al. [70] fed piglets with 0.9% or 1.8% potassium diformate, and observed a reduction in pH, the number of coliforms and streptococci in the stomach and proximal colon, and a reduction in coliforms in the colon, while no difference in lactobacilli was detected in the segments of the GIT tested. These results have also been confirmed by Kirchgessner et al. [76] and Gabert et al. [77]. *Clostridium perfringens* seems slightly affected by the addition of potassium diformate. In fact, its count was not affected by the addition of potassium diformate to a weaner diet [70]. Conversely to what was reported by Fevrier et al. [70], the addition of potassium diformate to a weaner diet at a concentration of 1.8% did not significantly affect the pH along the GIT, but increased the concentration of formic acid in the stomach and small intestine [69]. Despite this, a decreased number of total anaerobic bacteria, lactic acid bacteria and yeasts were found along the GIT, while the counts of coliforms in the small and large intestine were numerically, but not significantly reduced [69]. 

Studies, based on blends of organic acids, which included formic acid have generally suggested a reduction in lactic acid bacteria in the large intestine [73,74], while there was no consensus regarding the level of coliforms [7,73,74].

In order to evaluate the potential effect of formic acid and its salts on contrasting pathogen colonization, some studies have used challenge models in weaned pigs. The pig challenge models most used were based on enterotoxigenic *Escherichia coli* (ETEC) and Salmonella challenges [78,79]. In general, formic acid and its salts showed promising results in protecting weaning pigs from these pathogens by modulation of the gastrointestinal commensal bacteria and a reduction in the pathogen level. The addition of free organic acids (75% formic acid and 25% propionic acid) on experimentally Salmonella and *E. coli*-infected weaned pigs reduced the pathogen concentration in the stomach [80]. Canibe et al. [81] tested the addition of formic acid and ammonium formate to liquid feed in weaned pigs challenged with *E. coli* K12 and observed a reduction in lactobacilli and Enterobacteriaceae and a higher number of yeasts. Consensus results have been reported for free and protected calcium formate in reducing the severity and the duration of diarrhea on weaning piglets, challenged with ETEC F4 and ETEC K99, due to the effectiveness of the acidifiers in reducing ETEC and *E. coli* shedding [13,52,67]. Finally, Li et al. [71] showed that 0.5% potassium diformate tended to decrease the *E. coli* population and increase the lactobacilli count, without having any effect on the GIT pH in weaned pigs infected with ETEC F4.

Moreover, there is evidence of the effect of formic acid supplementation on growing and finishing pigs. Supplementation with 1.2% of potassium diformate in the diet of growing-finishing pigs reduced the number of coliform bacteria in the duodenum, jejunum and rectum [16]. Furthermore, Øverland et al. [14] reported that formic acid had a stronger antibacterial effect on coliforms in the small intestine than benzoic or sorbic acids. Finally, a reduction in lactobacilli and enterococci was also observed by Øverland et al. [82]. 

Other than preventing gut dysbiosis in piglets, formic acid has been tested in growing-finishing pigs to prevent the risks related to food-borne pathogens. Supplementation of the diet of finishing pigs with a mix of 0.6% lactic acid and 0.6% formic acid for the last 14 weeks before slaughter can reduce the presence of *Salmonella spp.* in the mesenteric lymph nodes [83]. Moreover, the same authors reported a reduction in Salmonella seroprevalences in finishing pigs fed for 8–9 weeks with a diet supplemented with 0.4% lactic acid and 0.4% formic acid. 

#### The metagenomics Approach

In addition, several *in vivo* trials have demonstrated the modulation of specific important groups of bacteria; the NGS approach allowed more complete information to be obtained regarding the effect of organic acids on the intestinal microbiota composition. Despite the potential utility, to date, few studies using the NGS approach have been carried out [8,10,58,59,84] and they were all focused on weaned pigs (Appendix A). The comparison among these studies is difficult because they differ as to the chemical type of the organic acid, the part of the intestinal tract analysed and the age of the animals. From these studies, two tested the effect of formic acid, one under normal healthy conditions [8] and the other under ETEC F4 challenge conditions [84]. No consistent results were observed in the alpha diversity index. The supplementation with formic acid reduced the microbial diversity indices of the bacteria adherent to the small intestinal mucosa, regardless of the dose [8], while in the study of Ren et al. [84], no difference was associated with the acid supplementation. While, considering the beta diversity index, consistent results were observed in the two studies, suggesting that formic acid did not influence the structures of the microbiome in the small intestine. 

Researchers have hypothesized that acidifier diets favour the growth of beneficial bacteria (such as Lactobacilli) and reduced the harmful bacteria (such as *E. coli*) by lowering the stomach pH or by exerting a direct effect on the harmful bacteria. However, the *in vivo* studies based on the NGS approach resulted in a slight decrease in Lactobacilli. The family Lactobacillaceae and the genus *Lactobacillus* were reduced by formic acid [8] and encapsulated benzoic acid [10]. These observations were in accordance with those obtained in previous *in vivo* studies, based on the culture-dependent method [53,66,69,73]. Furthermore, according to the study of Luise et al. [8], it can be observed that *Lactobacillus* was affected by the formic acid, and the additional genera belonging to lactobacilli (LAB) *sensu stricto*, including *Gemella* and *Streptococcus* genera, [85] were also reduced. The reduction in LAB in *sensu stricto* could have been influenced by the prolonged period of administration of formic acid (for 42 days from 7 days post-weaning) to weaners, which in turn, led to an adaptation of other commensal bacteria to the lower pH environment. 

Overall, the limited data make it difficult to explain the effect of formic acid on the intestinal microbiota, and additional studies are necessary. Furthermore, the few available data are focused on post-weaning piglets due to the potential beneficial effect of formic acid in this particular phase of life. However, potential promising results could also be expected regarding its use in growing-finishing pigs and in gestation sows. In-depth investigation of the effect of formic acid and its salts on the intestinal microbiota of growing-finishing pigs could, at least in part, explain the increase in growth performance associated to acidifier supply, given that intestinal microbiota plays a key role in the modulation of growth performance [44,86]. 

The scientific reason for studying the effect of formic acid in modulating the gut microbiota in pregnant sows is related to the contribution of sow’s gut microbiota; 1) improving the gut health of sows per se; and 2) favoring a safer environmental microbiota which could, in turn, promote the growth of more balanced microbiota in the gut and respiratory tracts of newborns. However, no study has yet been carried out to specifically address this topic nor, at least, has it been found in a survey of the literature, albeit that the dietary supplementation, with a mixture of organic acids and formic acid, beginning 42 days before farrowing increases the fecal count of lactobacilli in sows at farrowing and, thereafter, that of pigs at weaning [87]. 

## 4. The Effect of Formic Acid and its Salts on Pig Performance

The exact mechanisms whereby organic acids and their salts enhance animal growth and performance are not completely clear. Nevertheless, the effects might be related to several mechanisms, including the reduction in gastric pH and the diet buffer capacity, increased pepsin activation, thereby improving the dietary efficiency and the digestibility of protein and amino acids [4,24]. Furthermore, organic acid anions can complex with some minerals including calcium, phosphorus, magnesium and zinc, improving their digestion and decreasing the excretion of supplemental minerals and nitrogen [88,89]. In relation to organic acid salts, Eidelsburger et al. [90] noted that, where organic acids can reduce the gastric pH, the addition of formate did not. In fact, Eidelsburger et al. [90] and Kirchgessner et al. [76] have reported that the improvement in growth performance, resulting from the dietary inclusion of organic acid salts, is more related to the effect on the microbial composition. Moreover, the modification of the intestinal microbiota composition, as previously described, contributed to a shift in the SCFA derivative from the bacterial fermentation of carbohydrates (mainly acetate, propionate and butyrate), which could be available for the host energy metabolism. Acetate can enter into the tricarboxylic acid (TCA) cycle or can be used for the synthesis of cholesterol, ketone bodies, and long-chain fatty acids; propionate takes part in gluconeogenesis in the liver, and butyrate is involved in the mitochondrial fatty acid oxidation by which the resulting acetyl-CoA can be used in the same way as acetate [91]. In addition, SCFAs can stimulate the intestinal cell proliferation [92] and have previously been reported to be associated with organic acid supplementation in diets, resulting in improved mucosal growth, motility and proliferation of the epithelial cell lining [24]. 

Various studies have reported the effects of formic acid and its salts on pig growth performance (Table 3). 

### 4.1. Post-Weaning Pigs

As reported previously, the use of formic acid and its salts has been intensively tested in post-weaning pigs with a successful effect on growth performance. However, the results are still controversial. The meta-analyses carried out by Partanen et al. [99], and Partanen and Mroz [4] showed that formic acid and acid-formate additions at doses from 44 to 46 mEq improved the ADG, the feed conversion ratio (FCR) and the feed intake in weaning pigs [4,99], even if some studies did not report significant improvements regarding growth performance [19,93], and others have evidenced a dependent response according to the duration of the supplementation [8]. Considering potassium diformate utilization, several studies have reported an improvement in growth performance [12,96,97]. However, no significant effect on the average daily feed intake (ADFI), the ADG or FCR was observed by [69]. Differences in the administration of potassium diformate and potassium formate, between the pre-starter and the starter phase, have been reported by Htoo and Molares [11]. In fact, these authors did not observe any effect on the ADG, FCR, and body weight (BW) with the addition of organic acids during the pre-starter phase (days 0–14), while from 15 to 35 days of age and the overall 35 d periods, pigs supplemented with potassium diformate or potassium formate improved the ADG and ameliorated the FCR, compared with a control diet [11].

Fewer studies have been carried out regarding the use of Ca-formate and Ca/Na-formate as supplements, and the results for Ca-formate showed an increase in the FCR and the ADG [13] (Table 3). 

Regarding the use of blends of organic acids, the study of Walsh et al. [38] reported a negative influence on the ADFI and the pig growth rate when two organic acid blends (fumaric, lactic, propionic, citric and benzoic acids or phosphoric, fumaric, lactic and citric acids) were added to the diet. In pigs weaned on day 26, the daily live weight gain (DLWG) was enhanced by a mixture of formic acid (in a diatomaceous earth carrier), propionic acid, and potassium sorbate. Conversely, the growth performance of piglets weaned on day 36 improved by a mixture of formic acid, propionic acid and sodium benzoate, while in the same study, the FCR was not affected by weaning ages, but decreased in groups fed with experimental diets [34]. The study by Partanen et al. [34] pointed out some evidence regarding the need to design specific blends of additives, based on formic acid, when weaning ages differed. 

A more recent study has highlighted the benefits of MUFA (caprylic C8 and capric C10) together with formic and propionic acids in the piglet diet [100]. The authors highlighted that the use of SCFAs + MUFAs significantly and more rapidly increased the body mass with respect to the non-supplemented pigs, and to the formic and propionic diets pre and post-weaning. The FCR was affected only in the period from 35 to 56 days of age, and between the control group and the group fed with formic, propionic and caprylic acid [100]. Furthermore, Long et al. [7] demonstrated the efficacy of a blend of free and buffered SCFAs (mainly formic acid, acetic acid and propionic acid) combined with MCFAs as regards the FCR and the ADG of weaned piglets. 

The effects of formic acid and formate have also been studied in challenged animals infected with intestinal pathogens and discordant results have been reported regarding growth performance. A diet supplemented with free Ca-formate (1.2%) improved post-challenge BW, FI, gain-feed ratio (G:F) and the villous height in the small intestine in weaning pigs orally infected with *E. coli* K88 [13]. Conversely, after a challenge with *E. coli* K99, the addition of Ca-formate (1.8%) to the diet did not improve growth and reduced the villous height in the jejunum of piglets [67]. In both trials, the piglets were weaned at 21 days of age. However, the divergent effect of Ca–formate may be related to the different *E. coli* strains used for the challenge. 

In addition, in piglets infected with two pathogens, *E. coli* KCTC 2571 and *S. typhimurium*, a 0.4% acidifier blend diet with 17.2% formic acid, 4.1% propionic acid, 10.2% lactic acid, 9.5% phosphoric acid and silicon dioxide (SiO2) 34.0% was not effective in maintaining ADG, ADFI and the G:F [101]. These results partially confirmed the data reported by Torrallardona et al. [67], even if differences regarding the dose of acidifiers were evident. 

It has been reported that the reduction in stomach pH resulting from the organic acid addition to the diet of weaning pigs is one of the most beneficial effects related to improved growth performance. The maintenance of a low pH is a key element in host health and growth, and is dependent on the phase of life. In pigs, during suckling, the pH is regulated only by the lactic acid produced from lactose fermentation, while during early post-weaning, the lack of a lactose substrate and the introduction of solid feed having a high buffering capacity raises the pH [102]. In addition to the positive effect of organic acid on the pH, the addition of acidifiers could negatively affect the morphology and functionality of the stomach; in fact, free Ca-formate reduced the number of hydrochloric acid (HCl)-secreting parietal cells and H+/K+ ATPase gene expression in the fundic mucosa [103]. Therefore, particular attention should be paid to the dose and duration of the acidifier supplementation.

### 4.2. Growing and Finishing Pigs

In general, it has been reported that beneficial effects regarding growth performance in response to organic acids and their salts is more promising in growing-finishing pigs than in weaning pigs [4]. In that case, the organic acid performance-enhancing effect is largely attributed to the modulation of the microbiota [55] and, of the acidifiers, diets with formic acid and its salts have shown the best results [4]. Growth performance responses to the inclusion of formic acid and its salts can, however, be variable in individual experiments, depending on how long the acidifiers have been used as supplements [15]. 

Furthermore, a trend of an increased G:F ratio and ADG has been observed in pigs fed a diet containing formic acid [17] or formic acid-ammonium formate [95], for the entire growing-finishing period rather than for the growing period only. Eisemann and Heugten [15] tested different concentrations of formic acid-ammonium-formate supplementation in nursery, growing and finishing pigs (range 1.2%–0.6% of the diet), and the data showed higher growth performance when the acidifiers were used as supplements during all the growing phases, rather than in just one period. On the other hand, the improvement of ADG and the FCR during the growing period and not during finishing period as shown by Siljander-Rasi et al. [94] using a diet supplemented with formic acid and by Øverland et al. [16] using a diet supplemented with 0.8% potassium diformate. Øverland et al. [16], Stukelj et al. [98] reported that the addition of the acidifiers had no influence on growth performance when adding Ca/Na-formate and a mixed blend of acids, respectively; Øverland et al. [82] noted a tendency of the FCR to increase when using a diet with 1% formic acid. 

The divergent results could be due to different factors, such as the buffering capacity of the diets [104], or the characteristics of each feedstuff used. It is well known that cereals and cereal by-products have a low buffering capacity, and protein feedstuffs and minerals have intermediate and high buffering capacities, respectively. 

### 4.3. Sows 

There are few studies regarding the use of formic acid and its salts in gestating or lactating diets. However, there are two main scientific reasons of affirming the potentially positive effect of formic acid and its salts on sow reproductive performance. The first reason is the potential beneficial effect on the intestinal microbiota as reported in the previous section, and the second is related to the relevance of formate in the one-carbon metabolism in animals. In fact, formate is a constituent of 10-formyl-tetrahydrofolate which is mainly generated from serine in the mitochondria or by formate in the cytosol; 10-formyl-tetrahydrofolate is a source for purine and thymidylate synthesis [105]. Furthermore, starting from this compound, the methyl cycle is replenished, and the epigenetic regulation of DNA is regulated by DNA- and histone-methyltransferases. These regulations are particularly important in the fetal phase where the tissue differentiation is nutritionally dependent. Recent studies on rats [106] and sheep [107] have demonstrated that formate concentration peaks in the fetal circulation, compared with maternal plasma, during the mid-third gestation phase (sheep) and also continuing until birth (rat). The elevated values of related amino acids (serine, methionine and glycine) in the fetal plasma, then in the maternal plasma, may support the relevance of all these nutrients for the development of the fetus. However, once again, no studies regarding sows were found in the literature

Nevertheless, the few studies carried out on sows may promote interest in carrying out additional studies. Mroz et al. [108] tested the effects of formic acid supplementation on pregnant and lactating sows. The authors concluded that the addition of 1% formic acid did not affect the body mass in the total cycle (pregnancy and lactation), but increased the feed intake during lactation, and slightly improved litter size and piglet birth weight. In addition, formic acid supplementation showed anti-agalassia properties [108]. Øverland et al. [109] obtained comparable results when primiparous and multiparous sows were given feed supplemented with 0.8% or 1.2% potassium diformate in the gestation and lactation phases. In fact, increased individual piglet BW at birth and at weaning was observed at whatever the dose. A heavier BW at weaning was also observed when sows were fed sodium buffered formic acid (0.9% of the diet) during lactation [110]. Moreover, the supplemented sows reached the peaked of feed intake earlier than the control [110]. This agreed with the reduced weight and backfat loss observed by Lückstädt [111] in lactating sows supplemented with 0.8% potassium formate from pre-farrowing until weaning. 

#### Potential Risk

In addition to the observed beneficial effects of formic acid, and its salts on the growth performance of pigs and the reproduction performance of sows, a problem regarding the increase in bacterial resistance to an extreme acid condition arises and should be taken into account, particularly in the current context in which the attention to resistant bacteria has increased. The development of certain complex counteractive mechanisms of some microbial strains, such as Salmonella and *E. coli*, has been reported by Bearson et al. [112]. The adaptive acid tolerance response was found to be controlled by some mutations at the genes fcl (Fucose, FX-like), wecA (rfe), wecB (rffE) and waaG (rfaG). These mutations caused the loss of surface O-polysaccharide, O-polysaccharide plus enterobacterial common antigen (ECA) and ECA, which are crucial in the resistance and adaptation to acids in *Salmonella spp*. and *E. coli* [113,114,115]. These authors showed that the adaptive acid tolerance mechanism involved two distinct responses: Pre-challenge adaptation and transient adaptation, which occur during low pH challenge. The constant exposition to antimicrobial substances could interact with the bacterial community and could consequently create the predominance of resistant strains capable of competing with sensitive strains. Even if there is some evidence which excludes the occurrence of resistance of the bacteria to formic acid, few data are available to support the “resistance hypothesis” against acidifiers and/or other antimicrobial additives, and additional research is necessary. 

## 5. Conclusions

A review of the literature has shown that formic acid and its salts are potential performance enhancers in weaned piglets, finishing pigs and pregnant sows. According to the review of the literature and depending on the buffering capacity of the diet a dose of 8–10 g/kg and 10–12 g/kg of feed for formic acid, and sodium formate, respectively for weaning and growing pigs could be recommended. While, no sufficient literature is available to suggest a dose with respect to sows. A gradual reduction in the dose of formic acid supplementation to the diet would benefit animals during the transition to a feed without acidifier supplementation.

Furthermore, a review of the literature evidences that the effect of formic acid and its salts promotes pig performance by modulating the intestinal microbiota has still to be elucidated. In fact, the literature has reported contrasting results in intestinal bacterial modulation according to dose, duration, chemical form, intestinal tract, type of diet and age of the animal. 

The potential of using the NGS approach to elucidate this lack of knowledge should be considered. Furthermore, special attention should be paid to studying the modulation of the microbiome in:

(1) A targeted tract of the gut (e.g., no information is reported for the stomach in which formic acid exerts its primary effect); 

(2) pregnant sows due to the importance of the microbiome in the health of sows and in the early colonization of their offspring; the implication of the epigenetic effect of formic acid on offspring development, including the interplay with the microbiota should also be studied.

Therefore, additional studies are necessary to elucidate the mechanisms of action of formic acid, and its salts, as feed additives in pig nutrition.

## Figures and Tables

**Table 1 animals-10-00887-t001:** Physicochemical properties of most common SCFAs and their salts.

Name ^1^	Registration Number/EC	Physical Form	Mol.wt/GE (MJ/Kg)	Dissociation Constant (pK_a_)	CR ^2^	Odour
Formic	E 236	Liquid	46.03/5.7	3.75	+++	Pungent
Acetic	E 260	Liquid	60.05/14.6	4.76	+++	Pungent
Propionic	1a297	Oily liquid	74.08/20.6	4.88	++	Pungent
Butyric	-	Oily liquid	88.12/24.8	4.82	+	Rancid
Lactic	E 260	Liquid	90.08/15.1	3.86	(+)	Sour milk
Sorbic	E 200	Solid	112.1/27.85	4.78	(+)	Mildly acrid
Fumaric	2b08025	Solid	116.1/11.5	3.02/4.38	0 to (+)	Odorless
Malic	E 296	Solid/Liquid	134.1/10.0	3.46/5.10	(+)	Apple
Citric	E 330	Solid	192.1/10.2	3.1/4.8/6.4	0 to ++	Odorless
Ca-formate	E 238	Solid	130.1/11.0		0	Neutral
Ca-lactate	E 327	Solid	308.3/30.0		0	Neutral
Ca-propionate	E 282	Solid	184.1/40.0		0	Neutral
K-diformate	1a237a	Solid	130.0/11.4		0	Neutral
Ca- butyrate	-	Solid	214.0/48.0		0	Rancid
Mg-citrate	-	Solid	214.4/10.0		0	Neutral

^1^ Monocarboxylic (1-6); dicarboxylic (7–8); tricarboxylic (9); organic salts (10–16); ^2^ CR = corrosiveness rate: high (+++), medium (++), low (+), negligible (0).

**Table 2 animals-10-00887-t002:** Summary of the effect of formic acid and its salts on the pig bacterial count in the gastrointestinal tract during the weaning phase.

Acidifier ^1^	Inclusion g/kg Diet	Initial Body Weight (BW)				Changes in Microbial Counts, log10 Colony-Forming Units (CFU)									References
			Stomach			Small Intestine			Large Intestine					Feces	
			Total	LAC	COLI	Total	LAC	COLI	Total	LAC	COLI	Total	LAC	COLI	
Formic acid	6	6					↑	↑		↓	↓				[66]
	12	6					↓	↓		↓	↓				[66]
	18	6					↓	↑		↓	↓				[66]
	24	6					↓	↓		↓	↓				[66]
	7						↓	↓							[53]
	14			-	↓		↓	↓							[53]
Ca-formate	18	7.5					↑	↑		↑	↓				[67]
Na-diformate	18	5.7					↓	↓		↓	↓				[68]
K-diformiate	18		↓	↓	-	↓	↓	-	↓	↓	-				[69]
	18	8			↓		-	↓							[70]
	5	7.8											↑	↓	[71]
Formic + Lactic acid	1 0+ 10	8.4	↓	↓	↓										[72]
Blend_1	4	6.7				↓	↓		-	↓					[73]
Blend_2	11	4.9					-	↑		↓	↑				[74]
Blend_3	21	4.9					-	↑		↓	↑				[74]
Blend_4	3												-	↓	[7]

^1^ Blend_1: 35% formic acid + 35% lactic acid + 20% citric acid + 10% sorbic acid; Blend_2: 23.1% formic acid + 13.3% lactic acid + 12.4% acetic acid + 0.76 phosphoric acid + 0.76 citric acid; Blend_3: 51.7% lactic acid + 29.0% formic acid + 17.0% acetic acid + 16.0% phosphoric acid + 0.85% citric acid; Blend_4: commercial blend of free and buffered short chain fatty acids (mainly formic acid, acetic acid and propionic acid) combined with MCFA.

**Table 3 animals-10-00887-t003:** Summary of the effects of formic acid and its salts on pig growth performance at different phase.

Acidifier	Quantity %	Phase	ADG	F:G	G:F	Reference
Formic acid	0.5	weaning	=	=		[93]
	0.2	weaning	=	=		[19]
	0.6	weaning	↑ ^1^	↓		[8]
	0.14	weaning	↑ ^1^	↓		[8]
	0.8	grower		↑		[94]
	0.8	finisher		=		[94]
	1	grower-finisher	↑			[16]
	1.8	grower-finisher	↑		↑	[17]
	1	grower	=	↑		[82]
	0.8	grower-finisher	↑	↑		[95]
	1	grower	↑	↑		[14]
	1	finisher	=	↑		[14]
Formic acid + ammonium formate	0.4	weaning	=	↓		[34]
	1.2	grower-finisher	↑	↑		[95]
	1	grower	↑		↑	[15]
K-diformate	1.8	weaning	↑	↑		[96]
	-	weaning	↑	↑		[97]
	1.8	weaning	=	=		[69]
	1.2	weaning	↑	↑		[11]
	1	weaning	↑	↑		[12]
	0.8	grower	↑		↑	[16]
K-formate	1.56	weaning	↑	↑		[11]
Ca-formate	1.2	weaning			↑	[13]
Ca/Na- formate	0.85	grower-finisher	↑			[16]
Blend_1	0.4	weaning	↑	=		[34]
Blend_2	0.4	weaning	↑	↓		[34]
Blend_3	0.3	grower-finisher	=			[98]

^1^ Results related to the first three weeks of administration; There were no significant differences after six weeks. Blend_1: 77.5% formic acid and ammonium formate, 20% propionic acid, and 2.5% potassium sorbate; Blend_2: 87.5% formic acid and ammonium formate, 20% propionic acid 10% propionic acid, and 2.5% sodium benzoate; Blend_3: 1% formic acid, 0.85% benzonic acid, 0.85% sorbic acid. ADG: average daily gain; F:G: feed to gain ratio; G:F: gain to feed ratio.

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
