# Peer review of "A Review of the Effect of Formic Acid and Its Salts on the Gastrointestinal Microbiota and Performance of Pigs"

_animals, 2020, doi:10.3390/ani10050887_

Round 1

Reviewer 1 Report

The reviewed paper contains some interesting information about influence of formic acid and its salts on the gastrointestinal microbiota and performance of pigs: piglets, sows and fatteners (grower and finisher). The title of the article is correct and is consistent with its content. The Abstract chapter is written correctly. The authors made an interesting analysis of the available literature regarding the use of formic acid and its salts in pig nutrition.

In conclusion, please provide quantitative recommendations for the use of formic acid and / or its salts in diets (g/kg DM of mixtures) for piglets, fattening pigs (grower and finisher) as well as pregnant sows and lactating sows.

In my opinion the manuscript can be published after making a few minor comments:

Line 12, add - especially piglets,

Line 17, instead of antibiotics - antimicrobial growth promoters,

Line 97, In Table 1, add a column with E symbols, e.g. formic acid - E236, etc.

Line 145, add - DM of,

Line 271, Please write Latin names in italics, e.g. Clostridium perfringens,

Line 284, Please write Latin names in italics, eg. Escherichia coli,

Author Response

The reviewed paper contains some interesting information about influence of formic acid and its salts on the gastrointestinal microbiota and performance of pigs: piglets, sows and fatteners (grower and finisher). The title of the article is correct and is consistent with its content. The Abstract chapter is written correctly. The authors made an interesting analysis of the available literature regarding the use of formic acid and its salts in pig nutrition.

In conclusion, please provide quantitative recommendations for the use of formic acid and / or its salts in diets (g/kg DM of mixtures) for piglets, fattening pigs (grower and finisher) as well as pregnant sows and lactating sows.

Au: Thanks for the comment. A recommended dose for formic acid and sodium formate, according to the data reported in the manuscript, was included in the conclusion both for weaning and growing pigs. Regarding sows, no sufficient data are available to propose a quantitative recommendation for the use of formic acid and / or its salts.

In my opinion the manuscript can be published after making a few minor comments:

Line 12, add - especially piglets

Au: thanks, we added it

Line 17, instead of antibiotics - antimicrobial growth promoters,

Au: we would prefer to keep antibiotics instead of antibiotics as growth promoter (AGPs) since from 2006 their use has been forbidden in some country including Europe and Australia. In fact, acidifiers are known as a promising feeding strategy to reduce the use of therapeutic antibiotics.

Line 97, In Table 1, add a column with E symbols, e.g. formic acid - E236, etc.

Au: thanks for the suggestion, we added the Registration numbers in the tables.

Line 145, add - DM of,

Au:  thanks for the comment. Actually the regulation does not refer on DM of feed but claims a dose of formic acid in a feed with a 12 % moisture. We revised the manuscript according to the regulation.

Line 271, Please write Latin names in italics, e.g. Clostridium perfringens,

Au: thanks, we revised it.

Line 284, Please write Latin names in italics, eg. Escherichia coli,

Au: thanks, we revised it.

Reviewer 2 Report

The manuscript entitled “A review of the effect of formic acid and its salts on the gastrointestinal microbiota and performance of pigs” aimed to evaluate the effects of formic acid and its salts on the performance and the gastrointestinal microbiota balance of pigs. The article is rich in content and comprehensive in description. The effects of formic acid and its salts on the production performance of pigs and the balance of gastrointestinal flora are well summarized, and according to the current research situation, the shortcomings and the future research direction are pointed out, which is of great guiding significance to the industry. I think there are only a few minor problems that need to be revised here, and then acceptable.

Line 83: “[HA] the concentration of the undissociated acid” should be “[HA] is the concentration of the undissociated acid”

Line 198-200: Regarding the reason of the increase in pH values in the contents of the small and large intestines of weaned piglets, is there any explanation in the literature?

Line 431: The issue of reference format.

Author Response

The manuscript entitled “A review of the effect of formic acid and its salts on the gastrointestinal microbiota and performance of pigs” aimed to evaluate the effects of formic acid and its salts on the performance and the gastrointestinal microbiota balance of pigs. The article is rich in content and comprehensive in description. The effects of formic acid and its salts on the production performance of pigs and the balance of gastrointestinal flora are well summarized, and according to the current research situation, the shortcomings and the future research direction are pointed out, which is of great guiding significance to the industry. I think there are only a few minor problems that need to be revised here, and then acceptable.

Line 83: “[HA] the concentration of the undissociated acid” should be “[HA] is the concentration of the undissociated acid”

Au: thanks, we revised it.

Line 198-200: Regarding the reason of the increase in pH values in the contents of the small and large intestines of weaned piglets, is there any explanation in the literature?

Au: Thank for the comment. The differences among the studies can be due to the effect of time between feeding the animals and the slaughtering. Indeed, according to the observation of Canibe et al. [17], the pH in the gastrointestinal tract, especially in the stomach and small intestine, can vary according to time after feeding at which samples were taken. For instance, the pH of the stomach can decrease from 4.5 at 0.5h after the animal meal to <3.0 after 8.5 hours. We included this aspect in the manuscript at L 204-207.

Line 431: The issue of reference format.

Au: thanks, we revise it.

Reviewer 3 Report

This is an excellent review on the use of formic acid in diets fed to pigs. The flow and target location of formic acid in the intestinal tract of pigs is the only unclear message. Does it mostly work in the stomach? Or is it more in the intestinal tract? L180 to 221 would be a great place to include this information.

L11 delete ‘since’

L54 delete ‘thereby and overcoming cultivation steps’

L141 replace ‘rules’ with ‘legislation’

L169 replace ‘represent’ with ‘is’

L180 to 221 it is unclear where the acid elicits its microbial modulation – stomach, small intestine, cecum, large intestine? It needs to be discussed how the acid would get to each region. Is it absorbed by the pig?

L527 this is the first mention that formic acids main target is the stomach. This needs to be made clear. It ties in with the above mentioned question.

Author Response

This is an excellent review on the use of formic acid in diets fed to pigs. The flow and target location of formic acid in the intestinal tract of pigs is the only unclear message. Does it mostly work in the stomach? Or is it more in the intestinal tract? L180 to 221 would be a great place to include this information.

Au: Thanks for the comment, we revised the section “The effect of formic acid and its salts on the gut microbial profile”, according to your suggestion at L 188-192. The main target site for free formic acid is the stomach. The gastric environment plays a crucial role for the bacteria selection and acts as an ecological barrier, thus it is fundamental in shaping the structure of the whole gut microbial community (Beasley et al., 2015). This is particularly relevant in young pigs that at weaning have a stomach pH that tends to 5 and an unstable intestinal microbial population along the intestinal tract.  Formic acid and its salts exert a direct and indirect actions on the gastric microbial population that can indirectly affect the microbial composition in the next tracts of the intestine.

According to the reviewed literature we observed that most of the data were focused on the small and large intestine instead of the stomach. It could probably due to the fact that small intestine, especially jejunum, is considered a target site for the evaluation of intestinal health and functionality, in particular for weaning pigs and the large intestine is more relevant for growing pigs. Thus we provided the available information on the effect of formic acids and its salts in all the gastrointestinal tracts.  We hope that this part is now clearer.

L11 delete ‘since’

Au: thanks, we revised it

L54 delete ‘thereby and overcoming cultivation steps’

Au: thanks, we revised it

L141 replace ‘rules’ with ‘legislation’

Au: thanks, we revised it

L169 replace ‘represent’ with ‘is’

Au: thanks, we revised it

L180 to 221 it is unclear where the acid elicits its microbial modulation – stomach, small intestine, cecum, large intestine? It needs to be discussed how the acid would get to each region. Is it absorbed by the pig?

Au: Thanks for the comment, we revised the section “The effect of formic acid and its salts on the gut microbial profile”, according to your suggestion as following:” The main target intestinal site for free formic acid and its salts is the stomach. The gastric environment plays a curtail role for the bacteria selection and acts as an ecological filter, thus it is fundamental in shaping the structure of the whole gut microbial community (Beasley et al., 2015). Formic acid and its salts exert indirect and direct actions on the gastric microbial population that can affect the microbial composition in the next tracts of the intestine”. Regarding the encapsulated form of formic acids, it can reach the small and large intestine and thus exert its function in these distal part of the intestine, however only a few studies tested the effect of encapsulated formic acid and the review was more focused on free formic acid. As previously reported in the first comment, data on the effect of formic acids and its salts in the gastric environment are very scarce and literature are more focused on the small and large intestine, thus we reported all the available information. We hope this part is clearer.

 L527 this is the first mention that formic acids main target is the stomach. This needs to be made clear. It ties in with the above mentioned question.

We included at L 188-192 an explanation that stomach is the primary gastrointestinal site of formic acid.